# Species specific differences in use of ANP32 proteins by influenza A virus

Jason S Long[1], Alewo Idoko-Akoh[2], Bhakti Mistry[1], Daniel Goldhill[1], Ecco Staller[1], Jocelyn Schreyer[1], Craig Ross[3], Steve Goodbourn[3], Holly Shelton[4], Michael A Skinner[1], Helen Sang[2], Michael J McGrew[2], Wendy Barclay[1]*

[1]Section of Molecular Virology, Imperial College London, London, United Kingdom; [2]The Roslin Institute, Royal (Dick) School of Veterinary Studies, University of Edinburgh, Midlothian, United Kingdom; [3]Institute for Infection and Immunity, St. George's, University of London, London, United Kingdom; [4]Influenza Viruses, The Pirbright Institute, Surrey, United Kingdom

**Abstract** Influenza A viruses (IAV) are subject to species barriers that prevent frequent zoonotic transmission and pandemics. One of these barriers is the poor activity of avian IAV polymerases in human cells. Differences between avian and mammalian ANP32 proteins underlie this host range barrier. Human ANP32A and ANP32B homologues both support function of human-adapted influenza polymerase but do not support efficient activity of avian IAV polymerase which requires avian ANP32A. We show here that the gene currently designated as avian ANP32B is evolutionarily distinct from mammalian ANP32B, and that chicken ANP32B does not support IAV polymerase activity even of human-adapted viruses. Consequently, IAV relies solely on chicken ANP32A to support its replication in chicken cells. Amino acids 129I and 130N, accounted for the inactivity of chicken ANP32B. Transfer of these residues to chicken ANP32A abolished support of IAV polymerase. Understanding ANP32 function will help develop antiviral strategies and aid the design of influenza virus resilient genome edited chickens.

DOI: https://doi.org/10.7554/eLife.45066.001

*For correspondence: w.barclay@imperial.ac.uk

## Introduction

Influenza A viruses (IAV) infect a wide range of host species but originate from wild birds. Zoonotic transmission from the avian reservoir is initially restricted by host specific species barriers. Infection of new host species requires the virus to bind to cell surface receptors, utilise foreign host cellular proteins whilst evading host restriction factors in order to replicate its genome, and finally transmit between individuals of the new host.

The negative sense RNA genome of influenza A virus (IAV) is replicated in the cell nucleus using a virally encoded RNA-dependent RNA polymerase, a heterotrimer composed of the polymerase basic 1 (PB1), polymerase basic 2 (PB2) and polymerase acidic (PA) proteins together with nucleoprotein (NP) that surrounds the viral RNA, forming the viral ribonucleoprotein complex (vRNP) (*Te Velthuis and Fodor, 2016*).

Crucially, the viral polymerase must co-opt host factors to carry out transcription and replication (*Te Velthuis and Fodor, 2016*). The PB2 subunit is a major determinant of the host restriction of the viral polymerase (*Almond, 1977*). Avian IAV polymerases typically contain a glutamic acid at position 627 of PB2, and mutation to a lysine, the typical residue at this position in mammalian-adapted PB2 (*Subbarao et al., 1993*), can adapt the avian polymerase to function efficiently in mammalian cells. We have suggested that the restriction of avian IAV polymerase is due to a species specific difference in host protein ANP32A (*Long et al., 2016*). Avian ANP32A proteins have a 33 amino acid insertion, lacking in mammals, and overexpression of chicken ANP32A (chANP32A) in human cells

**eLife digest** The influenza A virus pandemic of 1918 killed more people than the armed conflicts of World War 1. Like all other pandemic and seasonal influenza, this virus originated from bird viruses. In fact, avian influenza viruses continually threaten to spark new outbreaks in humans, but pandemics do not occur often. This is because these viruses must undergo several adaptations before they can replicate in and spread between people.

Viruses make new copies of themselves using the molecular machinery of the cells that they invade. The proteins that make up this machinery are often slightly different in different species, and so a virus that can replicate in cells of one species might not be able to do so when it invades a cell from another species. In 2016, researchers discovered that species differences in a cell protein called ANP32A pose a key barrier that avian influenza viruses have to overcome.

Now, Long et al. – including some of the researchers involved in the 2016 study – show that the avian influenza virus cannot replicate in chicken cells that lack ANP32A. Exploring closely related versions of the genes that produce ANP32A and its relative ANP32B in different species revealed the region of the protein that the virus relies on to support its replication. Long et al. speculate that by making a few small changes to the ANP32A gene in chickens, it might be possible to generate a gene-edited chicken that is resilient to influenza.

Close contact with poultry has led to hundreds of cases of 'bird 'flu' in South East Asia, many of which have been fatal. Moreover, if avian influenza viruses mutate further in an infected person, a new pandemic could begin. Stopping influenza viruses from replicating in chickens would prevent people from being exposed to these dangerous viruses, whilst also improving the welfare of the chickens.

DOI: https://doi.org/10.7554/eLife.45066.002

rescues efficient function of avian origin IAV polymerases (*Long et al., 2016*). Removal of the 33 amino acids from chANP32A prevents polymerase rescue, whilst conversely artificial insertion of the 33 amino acids into either huANP32A or B overcomes host restriction (*Long et al., 2016*). A naturally occurring splice variant of avian ANP32A lacks the first four amino acids of the 33 amino acid insertion, reducing the rescue efficiency of avian IAV polymerase in human cells (*Baker et al., 2018*). This may be due to the disruption of a SUMOylation interaction motif, shown to enhance chANP32A's interaction with IAV polymerase (*Domingues and Hale, 2017*). In human cells, both family members ANP32A and ANP32B (huANP32A/B) are utilised by human adapted IAV polymerases, and are thought to stimulate genome replication from the viral cRNA template, although the exact mechanism remains unclear (*Sugiyama et al., 2015*).

Here we demonstrate that the avian ANP32B clade is evolutionarily distinct from mammalian and other ANP32Bs. We demonstrate that two amino acids differences, N129 and D130, in the LRR5 domain of chANP32B render it unable to interact with and support IAV polymerase function. We used CRISPR/Cas9 to remove the exon encoding the 33 amino acid insertion from chANP32A or to knockout the entire protein in chicken cells. Edited cells that expressed the short chANP32A isoform lacking the additional 33 amino acids supported mammalian-adapted but not avian IAV polymerase activity. Cells completely lacking chANP32A did not support either mammalian or avian IAV polymerase activity and were resilient to IAV infection. These results suggest a strategy to engineer IAV resilience in poultry through genetic deletion or amino acid changes of the LRR domain of ANP32A protein.

## Results

### Phylogenetic analysis identifies that avian ANP32B is a paralog of mammalian ANP32B

To examine the relatedness of ANP32 proteins from different species, we constructed a phylogenetic tree using vertebrate ANP32 protein sequences using *Drosophila* mapmodulin protein as an outgroup. ANP32A and E homologues formed well-supported monophyletic clades which included multiple avian and mammalian species (*Figure 1*, *Figure 1—figure supplement 1*). Most vertebrate

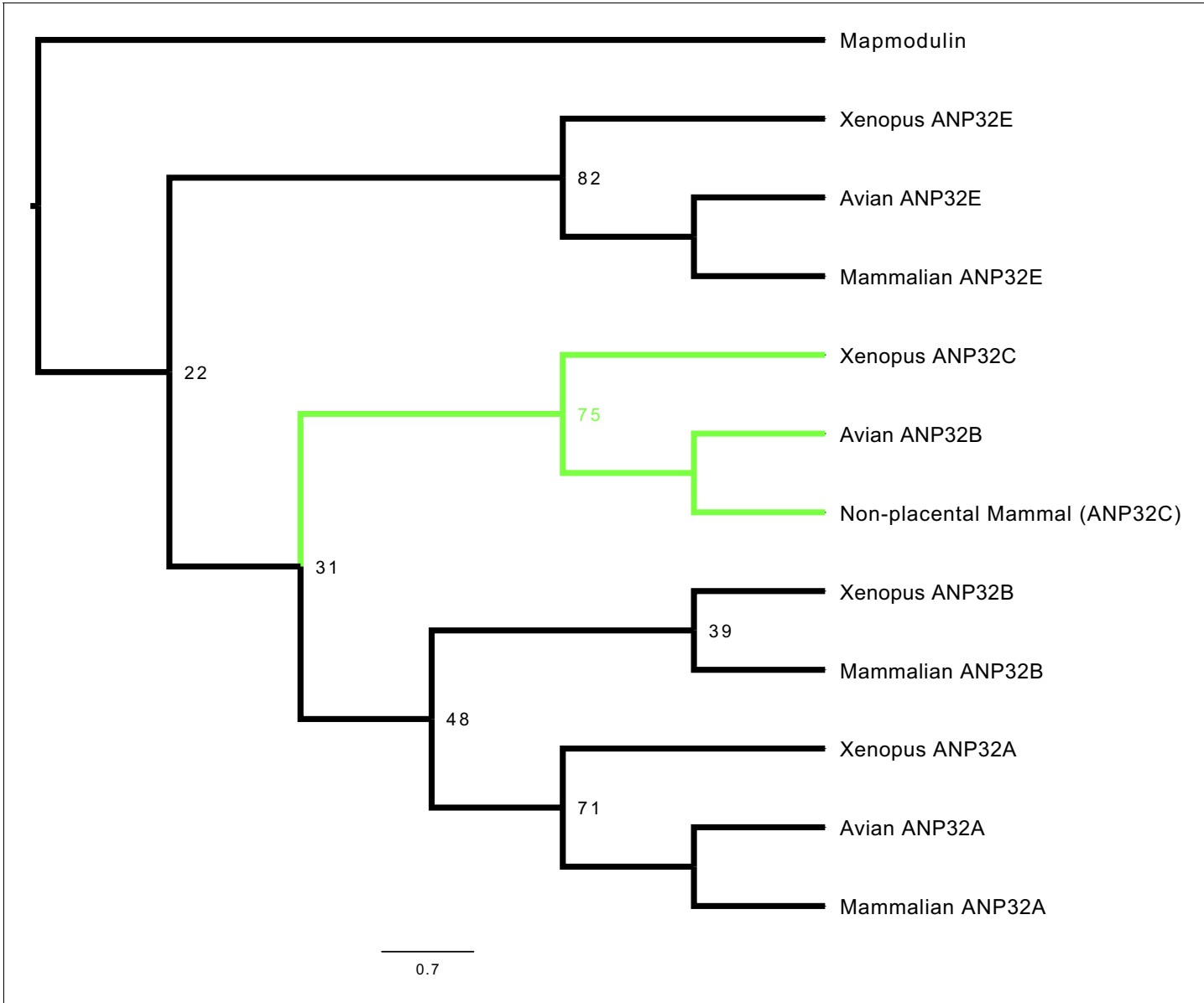

**Figure 1.** Phylogenetic and sequence analysis reveals avian ANP32B to be a paralog of mammalian ANP32B. The best maximum-likelihood tree was calculated from a set of ANP32 proteins with mapmodulin from *Drosophila melanogaster* as an outgroup using RAxML with 100 bootstraps. This figure is a cladogram showing the relationships between mammalian ANP32s, avian ANP32s and ANP32s from *Xenopus tropicalis*. Selected bootstrap values show the relationship between different ANP32 protein clades. Avian ANP32B clade is shown in green. The full tree is shown in *Figure 1—figure supplement 1*.

DOI: https://doi.org/10.7554/eLife.45066.003

The following figure supplements are available for figure 1:

**Figure supplement 1.** Phylogenetic and sequence analysis reveals avian ANP32B to be a paralog of mammalian ANP32B.
DOI: https://doi.org/10.7554/eLife.45066.004

**Figure supplement 2.** Synteny of ANP32 genes.
DOI: https://doi.org/10.7554/eLife.45066.005

ANP32B proteins formed a monophyletic clade but this clade did not include avian ANP32B proteins. Rather, avian ANP32B proteins were strongly supported as members of a distinct clade with ANP32C from *Xenopus* and unnamed predicted proteins from non-placental mammals. This suggests that avian ANP32B and mammalian ANP32B are paralogues: birds have lost the protein orthologous to human ANP32B and eutherian mammals have lost the protein orthologous to avian

ANP32B. Synteny provides further evidence to support the evolutionary relationship between avian ANP32B, *Xenopus* ANP32C, and the unnamed marsupial gene as they are all found adjacent to ZNF414 and MYO1F on their respective chromosomes (*Figure 1—figure supplement 2*). In humans, we found a short stretch of sequence between ZNF414 and MY01F which appears homologous to avian ANP32B (*Figure 1—figure supplement 2*). This provides further evidence that a functional gene orthologous to avian ANP32B has been lost in placental mammals.

## Chicken ANP32B does not support IAV polymerase activity

We and others have previously shown that both human ANP32A and B proteins support activity of a human-adapted IAV polymerase in human cells (*Long et al., 2016*; *Sugiyama et al., 2015*; *Watanabe et al., 2014*). Using CRISPR/Cas9, we generated human eHAP1 cells that lacked expression of both human ANP32A and ANP32B protein (Staller et al. *in review*). In WT eHAP1 cells, human-adapted IAV polymerase (PB2 627K), derived from an H5N1 virus A/turkey/England/50-92/1991 (50-92), was active, whereas the WT avian polymerase (PB2 627E) was not. Exogenous expression of C-terminally FLAG-tagged chANP32A could rescue the activity of avian IAV polymerase whereas expression of chANP32B-FLAG, which naturally lacks the 33 amino acid insertion, did not (*Figure 2a*). In double knockout cells, neither human-adapted nor avian-origin polymerase were active. Expression of chANP32A-FLAG rescued activity of both polymerases but expression of chicken ANP32B-FLAG rescued neither, despite confirmation of robust expression by western blot (*Figure 2b and c*). This suggests that chicken ANP32B is not functional for IAV polymerase and that the IAV polymerase activity relies on ANP32A in chicken cells. To confirm this in chicken cells, we used CRISPR/Cas9 gene editing to generate chicken DF-1 cells which lacked chANP32B but retained chANP32A expression (*Figure 2—figure supplement 1*). Wild type DF-1 cells had mRNAs for chANP32A, B and E (*Figure 2—figure supplement 1*) and supported activity of avian IAV polymerase bearing either PB2 627E or 627K. Overexpression of chANP32B-FLAG did not affect activity (*Figure 2d*). DF-1 bKO cells also supported activity of both polymerases and again, exogenous expression of chANP32B had no effect. Since chicken cells lacking expression of chANP32B did not demonstrate any loss of IAV polymerase activity compared to WT, this implied that chANP32B is not functional for IAV polymerase and that IAV polymerase uses solely ANP32 family member A in chicken cells.

## Chicken cells lacking intact ANP32A do not support avian IAV polymerase activity

To investigate the function of ANP32A in chicken cells we utilised a cell type that is more amenable to genome editing and clonal growth. Primordial germ cells (PGCs) are the lineage restricted stem cells which form the gametes in the adult animal. PGCs from the chicken embryo can be easily isolated and cultured indefinitely in defined medium conditions (*van de Lavoir et al., 2006*; *Whyte et al., 2015*). Chicken PGCs can be edited using artificial sequence-specific nucleases and subsequently used to generate genome edited offspring (*Park et al., 2014*; *Oishi et al., 2016*). Under appropriate in vitro conditions PGCs can acquire pluripotency and be subsequently differentiated into multiple cell types (*Matsui et al., 1992*; *Shim et al., 1997*; *Shamblott et al., 1998*; *Park and Han, 2000*). Chicken PGC cells were genome edited using CRISPR/Cas9 and a single guide RNA which generated chANP32A knock-out cells (aKO) containing a biallelic deletion of 8 nucleotides in exon 1. PGCs lacking the 33 amino acid insertion in chANP32A were generated using a pair of guide RNAs to remove exon five resulting in chicken cells with a mammalian-like ANP32A (Δ33) (*Figure 3a*). The precise deletions were confirmed by Sanger sequence analysis of subcloned PCR products from genomic DNA, and both found to be homozygous at both alleles (*Figure 3—figure supplement 1*). We differentiated the edited chicken PGCs into fibroblast-like cells using serum induction with the aim of generating cell lines to test avian IAV polymerase activity (*Figure 3—figure supplement 2*). The predicted alterations of ANP32A protein in these cells were confirmed by western blot analysis of the PGC-derived fibroblast cells (*Figure 3b*). WT, Δ33, and aKO and PGC-derived cell lines were tested for the functional effects of alteration or loss of chANP32A expression on IAV polymerase activity measured by reconstituted minigenome assay. Both avian (PB2 627E) and human-adapted polymerase (PB2 627K) were active in WT fibroblast cells (*Figure 3c*). Removal of the 33 amino acids from ANP32A resulted in restriction of the 627E polymerase but not the 627K

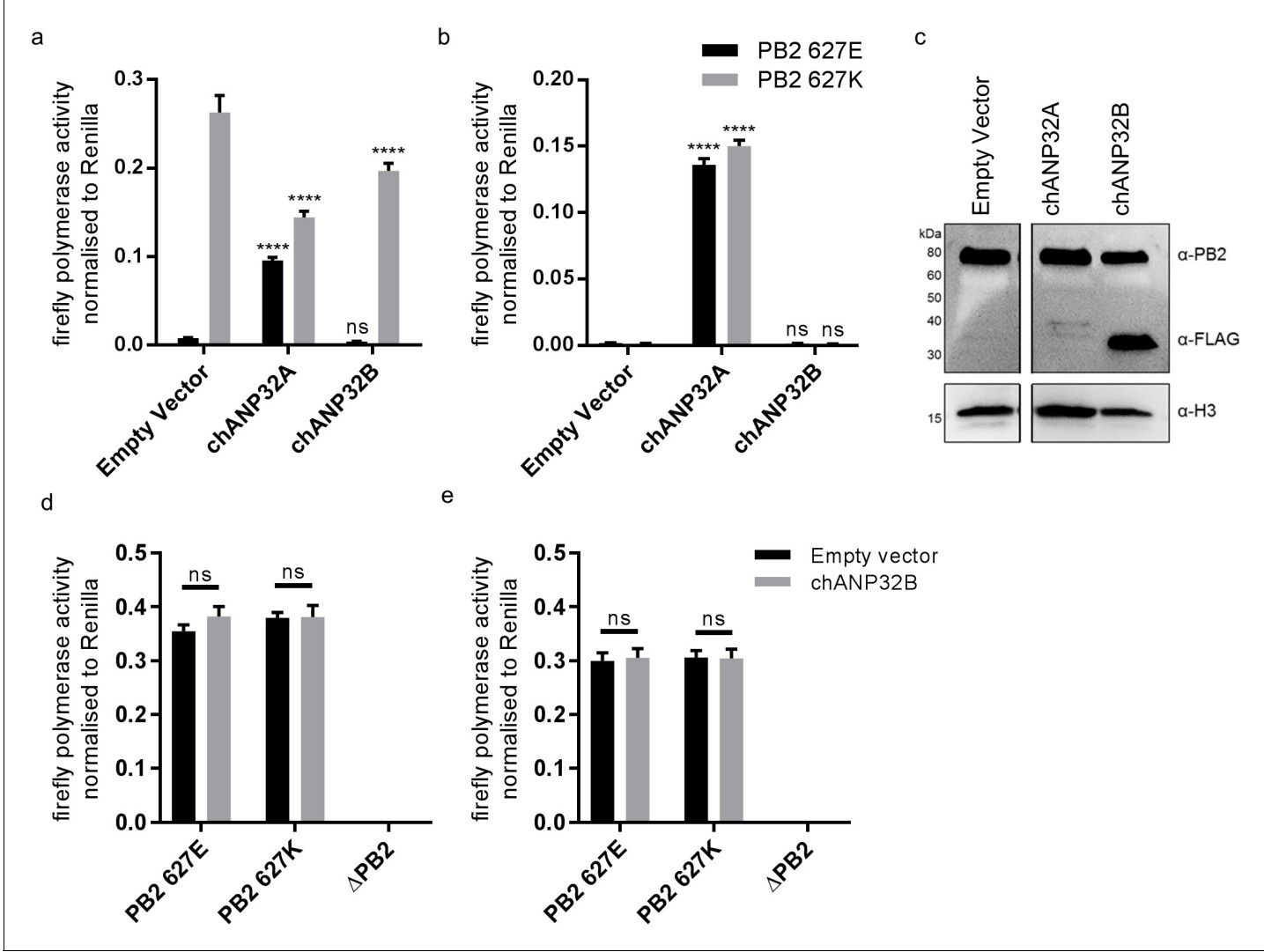

**Figure 2.** Chicken ANP32B is not functional for IAV polymerase. Cells were transfected with avian H5N1 50–92 polymerase (PB2 627E or 627K) together with NP, firefly minigenome reporter, *Renilla* expression control, either Empty vector (control) or ANP32 expression plasmid and incubated at 37°C for 24 hr. (a) Minigenome assay in human eHAP1 cells with co-expressed Empty vector, FLAG-tagged chANP32A or chANP32B. (b) Minigenome assay in double knockout (dKO) eHAP1 cells. (c) Western blot analysis of dKO eHAP1 cell minigenome assay confirming expression of PB2 and FLAG-tagged chANP32A and B. (d) Minigenome assay in WT DF-1 cells with either co-expressed Empty vector or chANP32B. (e) Minigenome assay in DF-1 ANP32B knockout (bKO) cells with either co-expressed Empty vector or chANP32B. Data shown are firefly activity normalised to *Renilla*, plotted as mean ± SEM (n = 3 biological replicates). Two-way ANOVA with Dunnet's multiple comparisons to Empty vector. ns = not significant, ****p<0.0001.

DOI: https://doi.org/10.7554/eLife.45066.006

The following figure supplement is available for figure 2:

**Figure supplement 1.** Sequence analysis of ANP32 in genome edited DF-1 chicken cells.

DOI: https://doi.org/10.7554/eLife.45066.007

polymerase, mirroring the avian IAV polymerase phenotype observed in mammalian cells (*Long et al., 2016*). Both polymerases were restricted in cells lacking chANP32A (aKO). Expression of exogenous chANP32A in Δ33 and aKO cells rescued avian IAV polymerase activity (*Figure 3d & e*) demonstrating the specificity of the genetic alterations. The lack of polymerase activity in the aKO PGC cell line supports the hypothesis that, in the absence of chANP32A, the remaining ANP family members including chANP32B or chANP32E could not support IAV polymerase activity in chicken cells, even though ANP32B and E mRNAs were readily detected in both DF-1 and PGC cells (*Figure 2—figure supplement 1* and *Figure 3—figure supplement 1*).

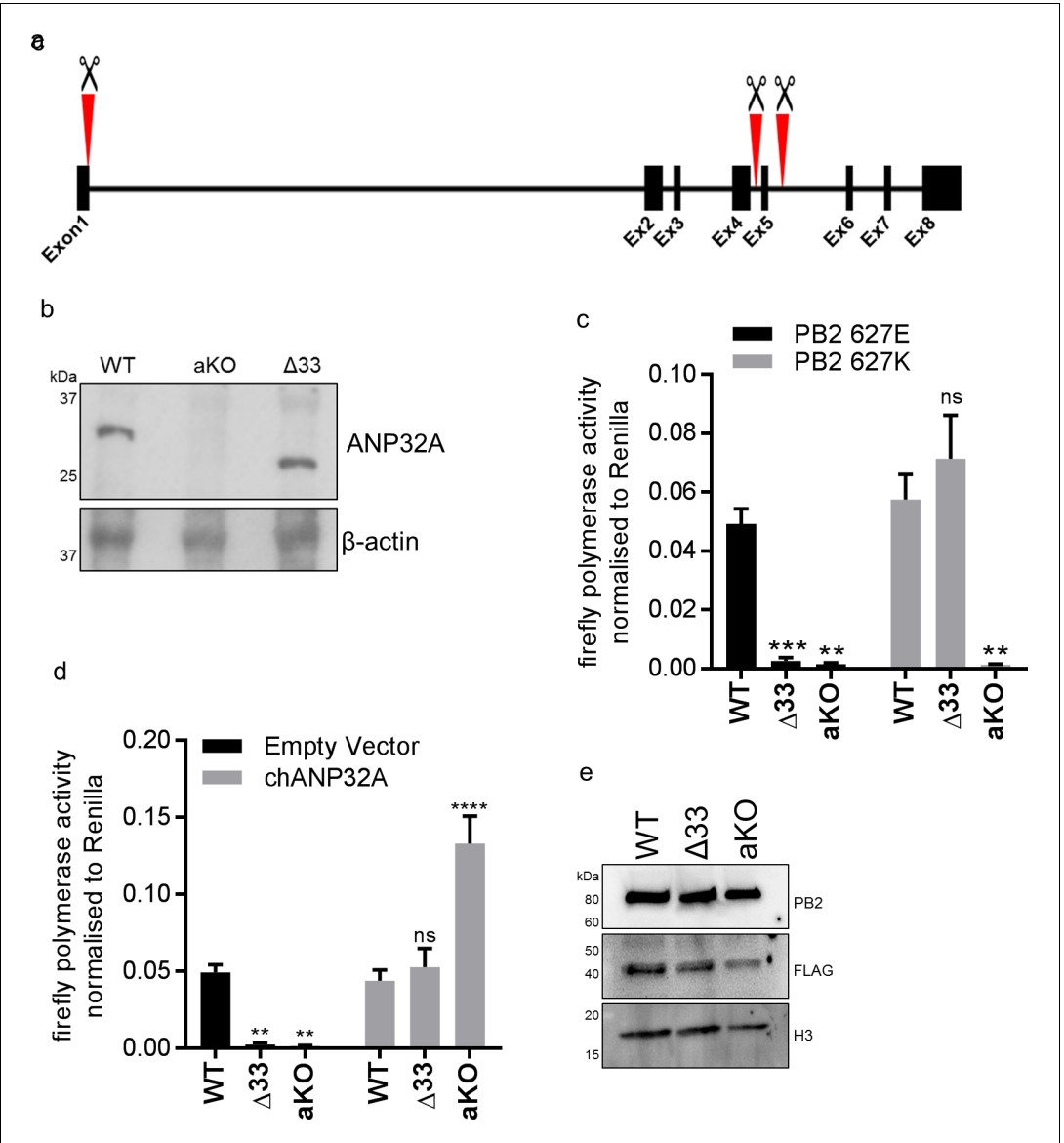

**Figure 3.** Chicken PGC derived fibroblast cells lacking ANP32A or the 33 amino acid insertion do not support avian IAV polymerase activity. (**a**) Schematic of CRISPR/Cas9 RNA guide targets used to generate aKO (exon1) and Δ33 (exon 5) PGC cell lines. (**b**) Western blot analysis of ANP32A and β-actin expression in WT, KO and Δ33 PGC-derived fibroblast cells. (**c**) Minigenome assay in WT, Δ33 or aKO PGC derived fibroblast cells with either PB2627E (black) or 627K (grey) polymerase derived from avian H5N1 50–92 virus. (**d**) Minigenome assay in WT, Δ33 or aKO cells with avian H5N1 50–92 PB2 627E polymerase co-transfected with Empty vector (black) or FLAG-tagged chANP32A (grey). (**e**) Western blot analysis of PB2, FLAG and Histone 3. Data shown are firefly activity normalised to *Renilla*, plotted as mean ± SEM (n = 3 biological replicates). Two-way ANOVA with Dunnet's multiple comparisons to WT. ns = not significant, **p<0.01, ***p<0.001, ****p<0.0001.

DOI: https://doi.org/10.7554/eLife.45066.008

The following figure supplements are available for figure 3:

**Figure supplement 1.** Sequence analysis of ANP32 in genome edited PGC chicken cells.
DOI: https://doi.org/10.7554/eLife.45066.009

**Figure supplement 2.** In vitro reprogramming of chicken PGCs into adherent fibroblast-like cells.
DOI: https://doi.org/10.7554/eLife.45066.010

## Functional differences between chicken ANP32A and ANP32B map to the LRR domain sequence

ANP32 proteins share a common domain organisation in which an N terminal domain consisting of 5 consecutive leucine rich repeats (LRR 1–5) is followed by a cap and central domain and a C terminal low complexity acidic region (LCAR). In avian ANP32A proteins (except some flightless birds) a sequence duplication, derived in part from nucleotides that encode 27 amino acids (149-175), has resulted in an additional exon and an insertion of up to 33 amino acids between the central domain and the LCAR (*Figure 4a*). We previously showed that insertion of the 33 amino acids from the central domain of chANP32A into the equivalent region of the human ANP32A or huANP32B proteins conferred the ability to rescue the activity of a restricted avian IAV polymerase in human cells. The equivalent 33 amino acid insertion into chANP32B (chANP32B$^{33}$) did not support avian IAV polymerase activity (*Figure 4b*). In order to ascertain the domains of chANP32B that rendered it non-functional for IAV polymerase activity, we generated chimeric constructs between human and chicken ANP32B. To measure the rescue of avian IAV polymerase in human 293 T cells, all chimeric constructs had the 33 amino acid sequence derived from chANP32A inserted between the LRR and LCAR domains. Western blot analysis and immunofluorescence confirmed that all chimeric constructs were expressed and localised to the cell nucleus as for the wild type ANP32 proteins. (*Figure 4b* and *Figure 4—figure supplement 1*). Swapping the LCAR domain of chANP32B into huANP32B$^{33}$ did not prevent the rescue of avian IAV polymerase (huANP32B$^{33}$$_{LCAR}$). Introduction of the central domain of chANP32B into huANP32B (huANP32B$^{33}$$_{CENT}$) significantly reduced rescue efficiency and swapping the LRR domain of chANP32B (huANP32B$^{33}$$_{LRR}$) rendered the protein non-functional to avian IAV polymerase (*Figure 4b*). By sequential swapping of each LRR repeat, the 5$^{th}$ LRR of chANP32B was found to be the domain that prevented rescue of avian IAV polymerase (*Figure 4b*). The fifth LRR contains five amino acid differences between human and chicken ANP32B, highlighted on the crystal structure of huANP32A, plus an additional one difference to chANP32A (*Figure 4d* and *Figure 4—figure supplement 2*). Swapping chANP32B's fifth LRR into chANP32A also prevented rescue of avian IAV polymerase activity in human cells (chANP32A$_{LRR5}$) (*Figure 4c*). Introduction of the single amino acid changes derived from the chANP32B LRR5 sequence into chANP32A revealed that mutations N129I and D130N significantly reduced the ability of chANP32A to rescue avian IAV polymerase activity in human cells (*Figure 4c*). Minigenome assays with co-expressed chANP32A or chANP32A$_{N129I}$ in aKO chicken fibroblast cells confirmed that the 129I mutation significantly reduced the ability of chANP32A to support avian-origin (PB2 627E) or human-adapted (PB2 627K) IAV polymerase activity (*Figure 4e*).

## Sequence of amino acids 149–175 of the central domain of chANP32A are required to support activity of both avian and human-adapted IAV polymerase

As chANP32A KO PGC-derived fibroblast cells did not support of IAV polymerase despite expressing chANP32B, we were able to use these cells to understand in more detail the sequences in chANP32A required for IAV polymerase activity. The results above showed that the 33 amino acid insertion, fifth LRR and central domain are important for the ability of chANP32A to support function of avian IAV polymerase. We performed the minigenome assay in aKO cells with polymerases containing either PB2 627E and 627K with co-expression of further chANP32 mutants including: chANP32A in which the 27 amino acids in the central domain preceding the 33 amino acid insertion were scrambled (chANP32A$_{scr149-175}$) or chANP32A with the 33 amino acid insertion scrambled (chANP32A$_{scr176-208}$) (*Figure 5a*). Both mutants were expressed and localised to the nucleus (*Figure 5c* and *Figure 4—figure supplement 1*). The first mutant, chANP32A$_{scr149-175}$, did not support either PB2 627E or 627K polymerase, suggesting the sequence of the central domain is important for function of IAV polymerase. The second mutant, chANP32A $_{scr176-208}$, only supported PB2 627K function, confirming that the sequence of the 33 amino acid insertion, not just the extended length is required for avian IAV polymerase (PB2 627E) (*Figure 5b*).

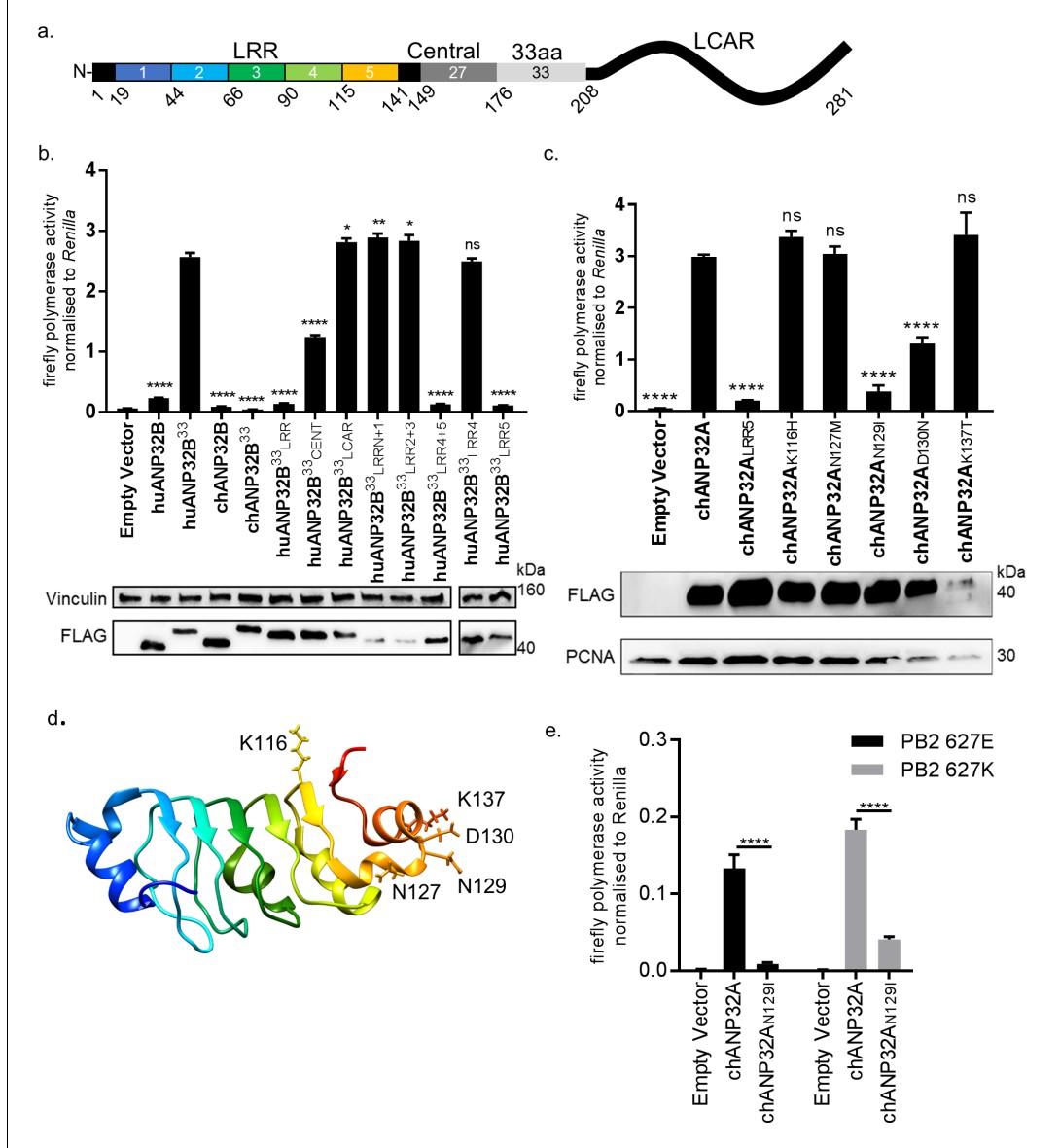

**Figure 4.** Lack of functional support for IAV polymerase by chicken ANP32B maps to differences in LRR5 domain. (a) Schematic of chicken ANP32A protein highlighting the different domains and LRR sequences (LRR 1–5). (b) Human 293 T cells were transfected with avian H5N1 50–92 polymerase (PB2 627E) together with NP, pHOM1-firefly minigenome reporter, *Renilla* expression control, either Empty vector or FLAG-tagged ANP32 expression plasmid and incubated at 37°C for 24 hr. Western blot analysis shown below (FLAG and Vinculin). (c) Minigenome assay in 293 T cells (PB2 627E) with FLAG-tagged WT or mutant chANP32A expression plasmids with associated western blot (FLAG and PCNA). (d) huANP32A crystal structure (PDB 4 × 05) with residues K116, N127, N129, D130 and K137 highlighted using UCSF Chimaera (*Pettersen et al., 2004*). (e) Minigenome assay of avian H5N1 50–92 polymerase with either PB2 627E or 627K in PGC-derived fibroblast aKO cells, together with co-expressed Empty vector, chANP32A or chANP32A$_{N129I}$. Data shown are firefly activity normalised to *Renilla*, plotted as mean ± SEM (n = 3 biological replicates). One-way ANOVA with Tukey's comparison to chANP32A (b and c) or two-way ANOVA with Dunnet's multiple comparisons to chANP32A (e). ns = not significant, *p<0.05, **p<0.01, ****p<0.0001.

DOI: https://doi.org/10.7554/eLife.45066.011

The following figure supplements are available for figure 4:

**Figure supplement 1.** Nuclear localisation of exogenously expressed ANP32 proteins in 293 T cells.
DOI: https://doi.org/10.7554/eLife.45066.012
**Figure supplement 2.** Avian ANP32B proteins share the I129 and N130 residues in LRR5.
DOI: https://doi.org/10.7554/eLife.45066.013

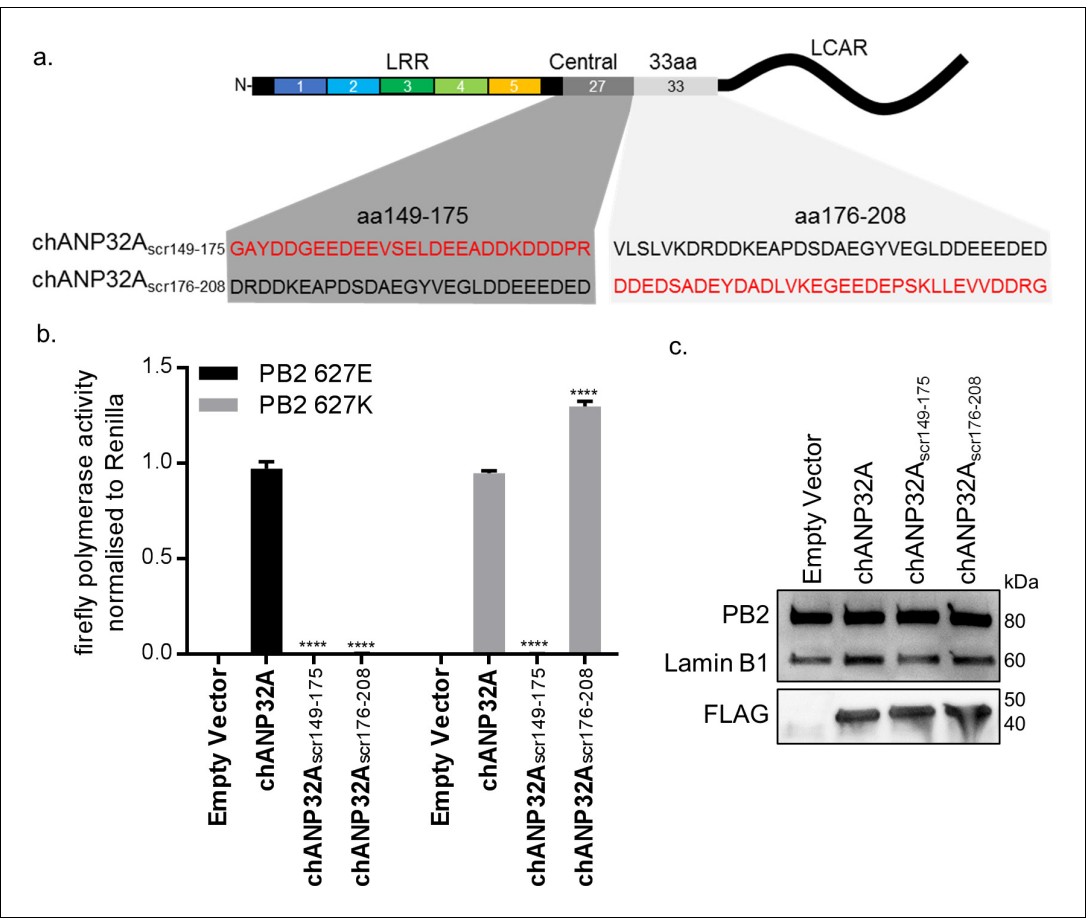

**Figure 5.** Sequence of amino acids 149–175 of the central domain of chANP32A are required to support activity of both avian and human-adapted IAV polymerase. (**a**) Schematic of chANP32A showing the sequence of amino acids in the central domain (149–175 or 33 amino acid insertion (176-208) and the randomly scrambled sequence in red. (**b**) Minigenome assay of avian H5N1 50–92 polymerase with either PB2 627E or 627K in PGC-derived fibroblast aKO cells with co-expressed Empty plasmid or FLAG-tagged WT chANP32A, chANP32A$_{scr149-175}$ or chANP32A$_{scr176-208}$ expression plasmids. (**c**) Western blot analysis of PB2 (627E), lamin B1 and FLAG. Data shown are firefly activity normalised to *Renilla*, plotted as mean ± SEM (n = 3 biological replicates). Two-way ANOVA with Dunnet's multiple comparisons to chANP32A. ns = not significant, ****p<0.0001.
DOI: https://doi.org/10.7554/eLife.45066.014

## A single amino acid difference between chANP32B and chANP32A abrogates binding of chANP32A to IAV polymerase

An interaction between ANP32A and IAV polymerase was demonstrated previously that is dependent on the presence of all three polymerase subunits (Mistry et al. *in preparation* & *Baker et al., 2018*; *Domingues and Hale, 2017*). To examine the interaction between IAV polymerase and chANP32 proteins we employed a split luciferase complementation assay as a quantitative measure of binding (*Munier et al., 2013*; *Cassonnet et al., 2011*). The C-terminus of the PB1 subunit of avian origin IAV polymerase was fused with one half of *gaussia* luciferase (PB1$^{luc1}$) and the C-terminus of chicken ANP32A or B with the second half (chANP32A$^{luc2}$ and chANP32B$^{luc2}$) (*Figure 6a*). Reconstitution of PB1$^{luc1}$, PB2 and PA together with chANP32A$^{luc2}$ in human 293 T cells gave a strong Normalised Luciferase Ratio (NLR) (*Figure 6—figure supplement 1*) with polymerases containing either PB2 627E or 627K (*Figure 6b*). Luciferase complementation was significantly less between polymerase and chANP32B$^{luc2}$, and even insertion of the 33 amino acids from chANP32A did not restore the signal (chANP32B$_{33}$$^{luc2}$) (*Figure 6b*). When chANP32A carried the single N129I mutation (chANP32A$_{N129I}$$^{luc2}$), luciferase complementation was reduced 22-fold for PB2 627E polymerase and 52-fold for PB2 627K polymerase (*Figure 6c* and *Figure 6—figure supplement 1*). These results suggest

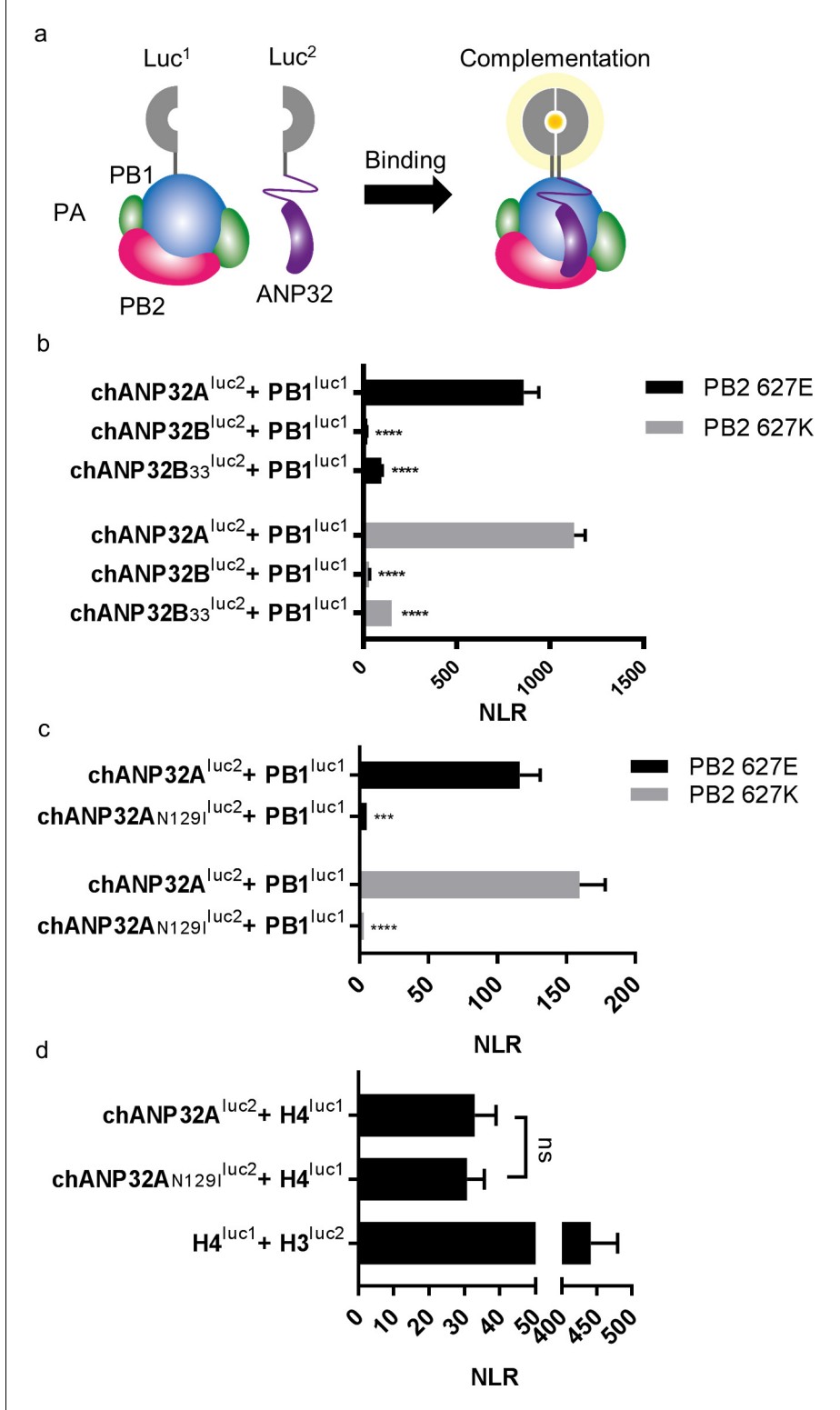

**Figure 6.** A single amino acid change (N129I) derived from chANP32B disrupts chANP32A support of influenza polymerase activity by abrogating binding to IAV polymerase. (a) Diagram of the split *Gaussia* luciferase system, demonstrating how ANP32 fused to luciferase fragment luc2 may bind to polymerase containing PB1 fused to luciferase fragment luc1 and complement full luciferase, which then reacts with substrate to generate a measurable bioluminescent signal. (b) Human 293 T cells were transfected with PB1 fused to luc1 (PB1$^{luc1}$), PB2 (627E or K), PA and either chANP32A, chANP32B or chANP32B$_{33}$ fused to luc2 (control wells were transfected with

*Figure 6 continued on next page*

*Figure 6 continued*

all components but with unfused PB1 and luc1 or chANP32 and luc2). (c) As (b) but with either chANP32A$^{luc2}$ or chANP32A$_{N129I}$$^{luc2}$. (d) 293 T cells transfected with either chANP32A$^{luc2}$ or chANP32A$_{N129I}$$^{luc2}$ and histone four fused to luc1 (or with unfused controls) or with H4luc1 and histone three fused to luc2. All data are Normalised Luciferase Ratio (n = 3 biological replicates) (*Figure 6—figure supplement 1*). One-way ANOVA (d) or two-way ANOVA with Dunnet's multiple comparisons to chANP32A (b and c). ns = not significant, ****p<0.0001.

DOI: https://doi.org/10.7554/eLife.45066.015

The following figure supplement is available for figure 6:

**Figure supplement 1.** Western blot analysis of split luciferase constructs.

DOI: https://doi.org/10.7554/eLife.45066.016

that the loss of support of polymerase function by chANP32A$_{N129I}$ was due to a disruption of binding to IAV polymerase.

ANP32A proteins bind to histones as part of their role in chromatin regulation (*Reilly et al., 2014*). To measure if the mutation N129I had any effect on this cellular interaction, we generated expression plasmids that encoded human histone four with luc1 fused to the C-terminus (H4$^{luc1}$) and histone 3.1 with luc2 fused to the C terminus (H3$^{luc2}$). As expected, H4$^{luc1}$ and H3$^{luc2}$ generated a strong NLR, reflecting their interaction in the nucleosome (*Luger et al., 1997*). The ability of chANP32A to bind histone four was not impaired by mutation N129I, suggesting chANP32$_{N129I}$ was not altered in this cellular role, despite abrogation of its support of IAV polymerase (*Figure 6d*).

## Viral replication is abrogated in chicken cells lacking ANP32A

The data above suggest that chANP32B cannot substitute for chANP32A in support of IAV polymerase in chicken cells. Since chicken cells that completely lack expression of chANP32A show no polymerase activity in the minigenome assay, they might be refractory to IAV infection. Multi-cycle growth kinetics of recombinant influenza A viruses were measured in WT and aKO PGC-derived fibroblast cells (*Figure 7*). To ensure robust infection, recombinant viruses were generated carrying H1N1 vaccine strain PR8 haemagglutinin (HA), neuraminidase (NA) and M genes; this also mitigated the risks of working with avian influenza viruses with novel antigenicity. Infectious titres of recombinant virus with internal genes of avian H5N1 virus 50–92 were not detected in the chicken cells lacking ANP32A infected at low MOI (*Figure 7a*). At higher MOI virus titres were significantly reduced compared to WT chicken cells, almost 325-fold less at 8 hr post infection and 16-fold less by 24 hr (*Figure 7c*). Similarly, at low MOI, a recombinant virus with internal genes from the H7N9 virus A/Anhui/1/2013, had limited virus growth in aKO cells but replicated efficiently in WT fibroblasts (*Figure 7b*). At the higher MOI, peak viral titres were 1365-fold less than in WT cells at 8 hr post infection and 100-fold less by 24 hr (*Figure 7d*). Since virus growth in aKO cells was observed at the higher MOI, we sequenced the PB2 gene of virus progeny to determine if this replication was due to adaptation in PB2. At 24 hr post infection the sequence of the PB2 gene from virus in supernatants of cells infected at high MOI was determined. Virus recovered from aKO and WT cells was found to be identical and contained no sequence changes compared with the inoculum, suggesting adaptation in PB2 was not required for the low level of replication seen in the aKO cells. In conclusion, PGC derived fibroblast cells lacking chANP32A were resilient to IAV replication, particularly at lower multiplicities of infection.

## Discussion

We show that avian origin IAV polymerases rely exclusively on chicken ANP32A family member for their replication, because they are unable to co-opt chicken ANP32B. We found avian ANP32B proteins formed a separate phylogenetic group from other ANP32Bs (*Figure 1* and *Figure 1—figure supplement 1*). Synteny demonstrated that an avian ANP32B homologue was present in coelacanth, amphibians and non-placental mammals as these loci were identical to the ANP32B locus in birds (*Figure 1—figure supplement 2*). A functional avian ANP32B homologue has been lost in placental mammals although a very small part of an ANP32 gene remained in humans (*Figure 1—figure supplement 2*). Human ANP32C is an intronless gene that is most closely related to ANP32A

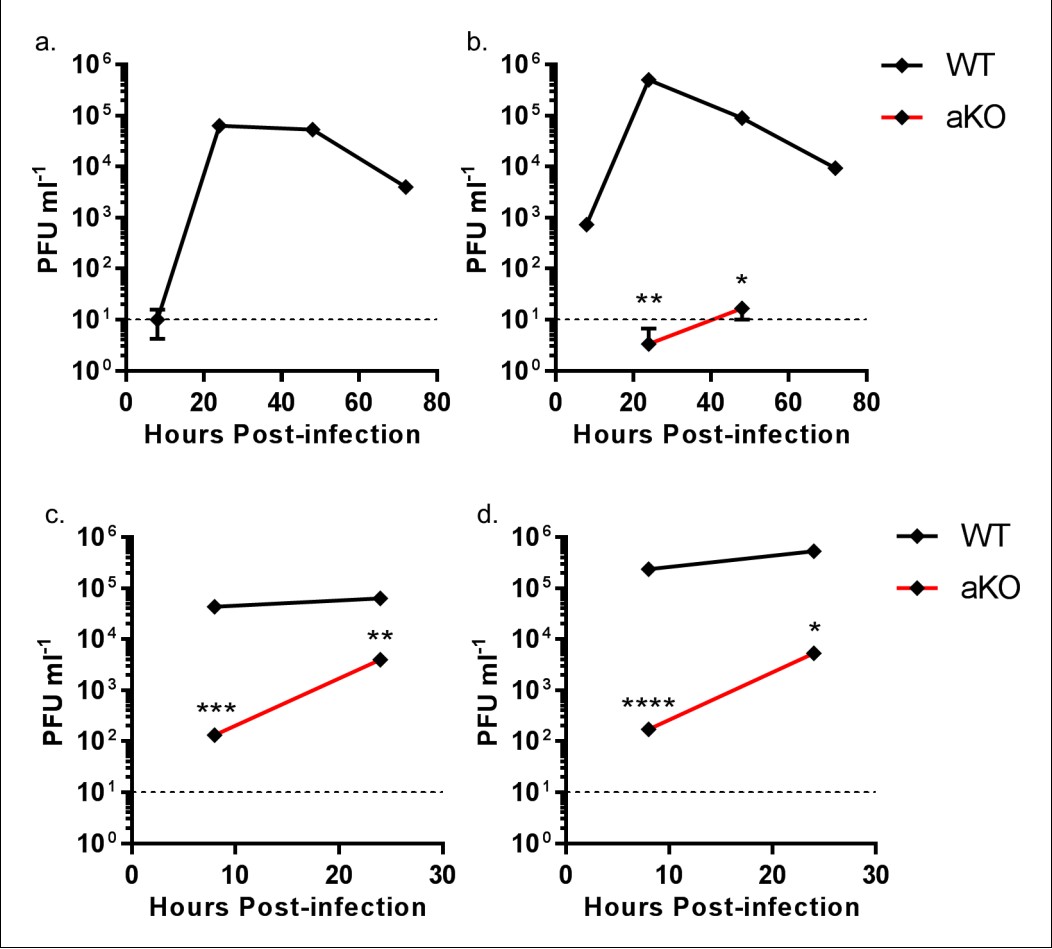

**Figure 7.** Viral replication is abrogated in chicken PGC fibroblast cells lacking ANP32A. WT (black lines) or aKO (red lines) PGC-derived fibroblast cells were infected with recombinant viruses (containing PR8 HA, NA and M genes and internal genes from either H5N1 50–92 or H7N9 Anhui), at an MOI of either 0.0001 (**a,b**) or 1.0 (**c,d**), incubated at 37°C in the presence of trypsin, cell supernatants harvested at described time-points and PFU ml$^{-1}$ measured by plaque assay on MDCK cells. (**a**) H5N1 50–92 (MOI 0.0001). b. H7N9 Anhui (MOI 0.0001). c. H5N1 50-−92 (MOI 1.0). d. H7N9 Anhui (MOI 1.0). vRNAs from supernatants 24 hr post infection MOI 1.0 (c and d) were extracted, PCR amplified and sequenced by Sanger sequencing. Limit of detection by plaque assay shown by dotted line (10PFU ml$^{-1}$). n = 3 biological replicates. Multiple t-tests with Holm-Sidak comparison. ns = not significant, *p<0.05, **p<0.01, ***p<0.001, ****p<0.0001.
DOI: https://doi.org/10.7554/eLife.45066.017

(*Reilly et al., 2014*) and is unrelated to the avian ANP32B clade. There was no evidence found of a mammalian ANP32B homologue in birds.

Chicken ANP32B could not support influenza polymerase function due to an amino acid difference in LRR5 at residue 129 that adversely affected the interaction between chANP32B and influenza polymerase (*Figure 6*). Other avian ANP32B proteins, including those of duck and turkey, carry isoleucine at residue 129 suggesting that our findings may also be applicable to other avian hosts (*Figure 4—figure supplement 2*). The replacement of the exposed polar residue, asparagine (N129) with the hydrophobic isoleucine (I) may have led to the disruption of a key electrostatic interaction between ANP32A and the virus polymerase complex. In addition to the residue 129I, the central domain (amino acids 141–175) of chANP32B also contributed to its poor efficiency at rescuing avian IAV polymerase function in human cells (*Figure 4*). This, together with the observation that scrambling amino acids 149–175 in chANP32A prevented both human-adapted and avian IAV polymerase function (*Figure 4*) suggests that LRR5 and the central domain of ANP32A are crucial to IAV polymerase function. Our finding that chANP32B is non-functional for IAV polymerase was recently

corroborated by another research group; this preliminary work also found that this phenotype mapped to residues 129I and 130N of chANP32B (*Zhang et al., 2019*). The observation that scrambling the 33-amino acid insertion prevented avian IAV polymerase rescue (*Figure 5*) is consistent with results from by Domingues and Hale and Baker and colleagues which showed that the SUMO Interaction motif (SIM)-like sequence present in the 33 amino acid insertion (VLSLV), was required for strong binding to both 627E and 627K polymerase and its deletion or mutation decreased its ability to support avian IAV polymerase activity in human cells (*Baker et al., 2018*; *Domingues and Hale, 2017*). Understanding the domains important to binding and function may help us understand the mechanism by which ANP32A or B support IAV polymerase which is still not fully elucidated (*Sugiyama et al., 2015*).

Chicken cells lacking ANP32A did not support activity of avian or human-adapted IAV polymerase in minigenome assay (*Figure 2*). However, at higher MOI, virus replication was observed in aKO cells, although still significantly lower than in WT PGC cells (*Figure 7c&d*). This implies some IAV polymerase function, albeit inefficient, in the absence of chANP32A in the context of virus infection. Other viral products present during virus infection such as NEP may partly compensate for the block in replication in cells that lack chANP32A. Indeed NEP expression has been reported to rescue avian polymerase replication in human cells (*Mänz et al., 2012*). Nonetheless the significantly reduced level of virus replication observed in the chicken cells that lack chANP32A in vitro implies that in vivo, chickens that do not express ANP32A or express altered protein may be resilient to infection by IAV. It will be pertinent to investigate whether IAV can evolve to replicate in chicken cells that lack or express a mutated ANP32A although no adaptation of PB2 gene was observed here. The discrepancy between the lack of polymerase activity in the minigenome assay in absence of ANP32A yet limited replication observed using infectious virus in the same cells may ultimately reveal interesting insights about how ANP32A supports polymerase. The use of the PGC-derived chicken cells to investigate a host factor essential for virus raises the possibility of generating genome-edited chicken models resistant or resilient to infection. Chicken PGCs can be efficiently genome-edited to generate specific haplotypes (*Idoko-Akoh et al., 2018*). Our novel method of chicken PGC differentiation into fibroblast-like cells enabled robust testing of a defined genotype, and will permit future investigation of other host genetic factors identified through forward genetic screens and suspected to play important roles in virus infections (*Smith et al., 2015*; *Wang et al., 2014*). We demonstrated that a mutated chANP32A was able to bind histone 4, suggesting this cellular role may not be affected by amino acid change at 129. However, there are many roles attributed to ANP32A proteins in the cell, such as embryogenesis, and disruption of these functions may limit the ability to generate healthy gene-edited animals (*Reilly et al., 2014*; *Reilly et al., 2011*).

In summary, we provide evidence that specific domains of ANP32 proteins are important for the function of IAV polymerases and describe a lack of redundancy in the involvement of ANP32 family members to support IAV polymerase complex in chicken cells that is determined by the variation in ANP32 protein sequences. These data may aid in the design of novel small molecule inhibitors that disrupt the ANP32-polymerase interface and form the basis of a potential pathway for the generation of influenza virus resilient animals.

# Materials and methods

**Key resources table**

| Reagent type (species) or resource | Designation | Source or reference | Identifiers | Additional information |
|---|---|---|---|---|
| Genetic reagent (*G. gallus*) | GFP[+]/Hyline cross | Roslin Institute | | Fertile heterozygous eggs for PGC derivations (*Pettersen et al., 2004*) |
| Primary cells (*G. gallus*) | Primordial Germ Cells (PGCs) | *G. gallus* GFP[+]/Hyline cross, Roslin Institute (This study) | | |

*Continued on next page*

*Continued*

| Reagent type (species) or resource | Designation | Source or reference | Identifiers | Additional information |
|---|---|---|---|---|
| Cell line (*G. gallus*) | DF-1 fibroblasts | American Type Culture Collection | CRL-12203; RRID:CVCL_0570 | |
| Cell line (*C. sapiens*) | MDCK | American Type Culture Collection | CCL- 34; RRID:CVCL_0422 | |
| Cell line (*H. sapiens*) | 293T | American Type Culture Collection | CRL-3216; RRID:CVCL_0063 | |
| Cell line (*H. sapiens*) | eHAP1 | Horizon Discovery | C669 | |
| Antibody | rabbit polyclonal α-ANP32A | Sigma-Aldrich | AV40203; RRID:AB_1844874 | Dilution 1:500-1:1000 |
| Antibody | mouse monoclonal α-β-actin | Sigma-Aldrich | A2228; RRID:AB_476697 | Dilution 1:1000 |
| Antibody | mouse monoclonal α-FLAG | Sigma-Aldrich | F1804; RRID:AB_262044 | Dilution 1:1000 (WB), 1:300 (IF) |
| Antibody | mouse monoclonal α-Lamin B1 | Merck | MAB5492; RRID:AB_2085944 | Dilution 1:1000 |
| Antibody | mouse monoclonal α-PCNA | Santa Cruz | sc-25280, RRID:AB_628109 | Dilution 1:1000 |
| Antibody | rabbit polyclonal α-Histone 3 | Abcam | AB1791; RRID:AB_302613 | Dilution 1:2000 |
| Antibody | rabbit monoclonal α-vinculin | Abcam | AB129002; RRID:AB_11144129 | Dilution 1:1000 |
| Antibody | rabbit polyclonal α-Gaussia Luc | NEB | E80235 | Dilution 1:2000 |
| Antibody | rabbit polyclonal α-PB1 | Invitrogen | PA5-34914; RRID:AB_2552264 | Dilution 1:2000 |
| Antibody | rabbit polyclonal α-PB2 | GeneTex | GTX125926; RRID:AB_11162999 | Dilution 1:2000 |
| Antibody | goat polyclonal anti-rabbit HRP | CST | 7074 | Dilution 1:2000 |
| Antibody | Horse polyclonal anti-mouse HRP | CST | 7076 | Dilution 1:2000 |
| Antibody | Sheep polyclonal α-rabbit HRP | Merck | AP510P | Dilution 1:20000 |
| Antibody | goat polyclonal α-mouse HRP | AbD Serotec | STAR117P; RRID:AB_323839 | Dilution 1:10000 |
| Antibody | goat polyconal α-mouse AlexaFluor-568 | Invitrogen | A11031; RRID:AB_144696 | Dilution 1:1000 |
| Recombinant DNA reagent | pGEM-T Easy vector | Promega | A1360 | |
| Recombinant DNA reagent | pSpCas9(BB)—2A-Puro PX459 V2.0 vector | Gift from Dr. Feng Zhang | RRID:Addgene_62988 | |
| Recombinant DNA reagent | pSpCas9n(BB)—2A-GFP PX461 vecotr | addgene | Plasmid 48140; RRID:Addgene_48140 | |
| Recombinant DNA reagent | pCAGGS vector | Belgium Co-ordinated Collections of Microorganisms (BCCM), University of Ghent, Belgium | | |

*Continued on next page*

*Continued*

| Reagent type (species) or resource | Designation | Source or reference | Identifiers | Additional information |
|---|---|---|---|---|
| Recombinant DNA reagent | H5N1 A/turkey/England/ 50-92/1991 poII plasmids | APHA, Weybridge, UK | | |
| Recombinant DNA reagent | H7N9 Anhui/1/2013 pHW2000 plasmids | Pirbright Institute, UK | | |
| Recombinant DNA reagent | H1N1 A/PR/8/34 (PR8) poII or pHW2000 plasmids | (*Neumann et al., 1999*) | | |
| Commercial assay or kit | RNeasy mini kit | Qiagen | 74106 | |
| Commercial assay or kit | Quick Start Bradford Protein Assay Kit | Biorad | 5000202 | |
| Commercial assay or kit | Dual-luciferase Reporter assay system | Promega | E1910 | |
| Commercial assay or kit | Renilla luciferase kit | Promega | E2810 | |
| Commercial assay or kit | QIAamp Viral RNA Mini Kit | Qiagen | 52906 | |
| Chemical compound, drug | Lipofectamine 3000 | Invitrogen | L3000008 | |
| Chemical compound, drug | Lipofectamine 2000 | Invitrogen | 11668019 | |
| Software, algorithm | Image J | ImageJ (http://imagej.nih.gov/ij/) | | |
| Software, algorithm | GraphPad Prism | GraphPad Prism (https://graphpad.com) | version 6 | |
| Software, algorithm | Geneious | Geneious https://www.geneious.com | R6 | |

## Animal use

The GFP$^+$ PGCs used in the experiments were obtained by crossing the Roslin Green (ubiquitous GFP) line of transgenic chickens with a flock of commercial Hyline layer hens maintained at the Roslin Institute to produce heterozygous fertile eggs for PGC derivations (*Pettersen et al., 2004*). Commercial and transgenic chicken lines were maintained and bred under UK Home Office License. All experiments were performed in accordance with relevant UK Home Office guidelines and regulations. The experimental protocol and studies were reviewed by the Roslin Institute Animal Welfare and Ethical Review Board (AWERB) Committee. Chickens for egg production were maintained under the HO code of practice (ISBN 9781474112390).

## Plasmid constructs

ANP32A guide RNAs (gRNA) were designed using CHOPCHOP gRNA web tool (http://chopchop.cbu.uib.no/) (*Montague et al., 2014*; *Labun et al., 2016*). gRNA 5'-CGGCCATGGACATGAAGAAA-3' targeting ANP32A exon1, and gRNAs: 5'-AGCTGGAAGCAATATGTACT-3' and 5'-CATTCCCCTCGCTCCTTCAA-3' targeting either side of exon 5 (Δ33 PGC cells) were cloned into pSpCas9(BB)−2A-Puro (pX459 v2.0; a gift from Dr. Feng Zhang) using Materials and methods described by previously (*Ran et al., 2013a*). For DF-1 ANP32B gRNAs, the guides 5'-TTCAGATGATGGGAAGATCG-3' and 5'-GGTTCTCAAAATCTGAAGAG-3' were cloned into the double 'nickase' vectors pSpCas9n(BB)−2A-GFP (pX461) and pSpCas9n(BB)−2A-Puro (pX462) respectively (*Ran et al., 2013a* ; *Ran et al., 2013b*).

*Gaussia* luc1 and luc2 were generated by gene synthesis (GeneArt, ThermoFisher) using the sequence previously described (**Cassonnet et al., 2011**). *Homo sapiens* Histone 4 (NP_003533.1) and 3.1 (NP_003520.1) were generated gene synthesis (GeneArt, ThermoFisher). Luc1 or luc2 were added to the C-termini of ANP32, PB1, H4 or H3.1 using the linker sequence, AAAGGGGSGGGGS, by overlapping PCR. The 33 amino acid insertion was added to huANP32B after residue 173 and to chANP32B after residue 181 (preserving an acid region before SIM motif [**Domingues and Hale, 2017**]). The LRR (amino acids 1–149), central domain (amino acids 150–175) or LCAR (amino acids 176–262) from chANP32B were swapped into huANP32B$_{33}$ to generate chimeric constructs. ANP32 constructs were made by overlapping PCR or by gene synthesis (GeneArt, ThermoFisher) with either a FLAG tagged fused to the C-terminus with a GSG linker or to mCherry with a GSGGGSGG linker.

## Cells and cell culture

Human embryonic kidney (293T) (ATCC) and Madin-Darby canine kidney (MDCK) cells (ATCC) were maintained in cell culture media (Dulbecco's modified Eagle's medium (DMEM; Invitrogen) supplemented with 10% fetal calf serum (FCS) (Biosera), 1% non-essential amino acids (NEAA) and with 1% penicillin-streptomycin (Invitrogen)) and maintained at 37°C in a 5% CO2 atmosphere. Human eHAP1 cells (Horizon Discovery) were cultured in Iscove's Modified Dulbecco's Medium (IMDM) supplemented with 10% fetal bovine serum (FBS), 1% NEAAs, and 1% penicillin/streptomycin. Chicken fibroblast (DF-1) (ATCC) cells were maintained in DF-1 cell culture media (DMEM supplemented with 10% FCS, 5% tryptose phosphate broth (Sigma-Aldrich), 1% NEAAs and 1% penicillin-streptomycin and maintained at 39°C in a 5% CO2 atmosphere. Cell line authentication: DF-1, eHAP1 and 293 T cells were authenticated by mRNA analysis confirming the relevant species. All continuous cell lines were routinely screened for mycoplasma contamination and were mycoplasma free.

## PGC, DF-1 and eHAP1 cell line generation

PGCs were derived and cultured in FAOT medium as previously described (**Whyte et al., 2015**). PGCs were transiently transfected with 1.5 µg of PX459 V2.0 vector using Lipofectamine 2000 (Invitrogen) and treated with puromycin as previously described (**Idoko-Akoh et al., 2018**). Subsequently, single cell cultures of puromycin-resistant cells were established to generate clonal populations for downstream experiments as previously described in **Idoko-Akoh et al. (2018)**. To identify an ANP32A Δ33 PGC cell line, PCR products were directly sequenced using PCR primers to analyse mutation genotypes of isolated single cell clones. To identify an ANP32A KO PGC cell line, PCR products were cloned into pGEM-T Easy vector (Promega) and sequenced using T7 promoter forward primer by Sanger sequencing. DF-1 cells were transfected with the described CRISPR/Cas9 constructs using Lipofectamine 2000 (Invitrogen) and subject to puromycin selection. Single cell clones were expanded and analysed by PCR of genomic DNA and Sanger sequencing using primers (5'-TTTTTGCTTACATCTGAGGGC-3', 5'-CCTCCGCAGTTATCAGGTTAGT-3') for ANP32A exon1, (5'-GCTCCCTGGTCTGCTAGTTAT-3', 5'-GGTCTACGCAACCACACATAC-3') for ANP32A exon five and (5'-CCCTTAAGGTGAGCACAGGG-3', 5'-AACATAGCACCACTCCCAGC-3') for ANP32B exon2. eHAP1 dKO cells were generated as described (Staller et al. *in review*).

## Differentiation of PGCs into adherent fibroblast-like cells (PGC derived fibroblasts)

PGCs were cultured in 500 µl of high calcium FAOT medium containing 1.8 mM CaCl$_2$ in fibronectin-coated wells (24-well plate) for 48 hr (**Figure 3—figure supplement 2**) (**Whyte et al., 2015**). Subsequently, PGCs were transferred into PGC fibroblast medium and then refreshed every 48 hr by removing and replacing with 300 µl of PGC fibroblast cell culture medium. Adherent fibroblast-like cells were observed within 72 hr. Cells were then refed every two days and split 1:4 every four days. PGC fibroblast cell cultures were expanded to 85–90% confluency in 24-well plates before using for transfection, infection or western blot analysis. PGC fibroblast cells were maintained in cell culture media (Knockout DMEM (10829018, Gibco) with 10% ES grade FBS (16141061, Invitrogen), 1% chicken serum (Biosera), 0.1% 100xNEAA (Gibco), 0.1% Pyruvate (11360070, Gibco), 0.1% 100xGlutamax (Gibco: 35050–038), 0.5 mg ml$^{-1}$ ovotransferin (C7786, Sigma)) and 1% penicillin-streptomycin at 37°C with 5% CO$_2$.

## Influenza A virus infection

Recombinant influenza A PR8 (A/PR/8/34 (H1N1)) 3:5 reassortant virus (PR8 HA, NA and M genes with PB1, PB2, PA, NP and NS genes from A/Anhui/1/13 (H7N9) was generated by reverse genetics at The Pirbright Institute, UK. Reverse genetics virus rescue was performed by transfection of Human Embryonic Kidney (HEK) 293 T cells (ATCC) with eight bi-directional pHW2000 plasmids containing the appropriate influenza A virus segments and co-culture in MDCK cells (ATCC) with addition of 2 µg ml$^{-1}$ of TPCK treated Trypsin (Sigma-Aldrich). Rescued viruses were passaged once in embryonated hen's eggs to generate working stocks. Recombinant PR8 3:5 reassortant 50–92 (A/turkey/England/50-92/1991 (H5N1) was described previously [*Long et al., 2013*]).

Virus was diluted in Knockout DMEM and incubated on PGC fibroblast cells for 1 hr at 37°C (MOI as indicated in the relevant figure legends) after which inoculum was removed and cells washed with PBS followed by MES buffer (pH 4, 37°C) for five mins and a further PBS wash. Infection media (Knockout DMEM (10829018, Gibco), 0.14% BSA and 1 µg ml$^{-1}$ TPCK trypsin (Sigma-Aldrich)) wash added and cells were incubated at 37°C. To ensure residual virus was removed, a 0 hr time point was taken. Cell supernatants were harvested and stored at −80°C. Infectious titres were determined by plaque assay on MDCK cells.

vRNA extraction from cell supernatants was performed using QIAamp Viral RNA Mini Kit (Qiagen 52906). First strand synthesis was performed using SuperScript IV reverse transcriptase with primer 5'-GCAGGTCAAATATATTCAATATGG-3'. cDNA was amplified using KOD Hot Start DNA polymerase (Merck 71086) using primers 5'-GCAGGTCAAATATATTCAATATGG-3' and 5'-GGTCGTTTTTAAACAATTCGAC-3' and the PCR product was sequenced by Sanger sequencing.

## Minigenome assay

Influenza polymerase activity was measured by use of a minigenome reporter which contains the firefly luciferase gene flanked by the non-coding regions of the influenza NS gene segment, transcribed from a species-specific polI plasmid with a mouse terminator sequence. The human and chicken polI minigenomes (pHOM1-Firefly and pCOM1-Firefly) are described previously (*Moncorgé et al., 2013*). pCAGGS expression plasmids encoding each polymerase component and NP for 50–92 (H5N1 A/Turkey/England/50–92/91) are described previously (*Long et al., 2013*). To measure influenza polymerase activity, 293 T cells were transfected in 48-well plates with pCAGGS plasmids encoding the PB1 (20 ng), PB2 (20 ng), PA (10 ng) and NP (40 ng) proteins, together with 20 ng species-specific minigenome reporter, either Empty pCAGGS or pCAGGS expressing ANP32 (50 ng) and, as an internal control, 10 ng Renilla luciferase expression plasmid (pCAGGS-Renilla), using Lipofectamine 3000 transfection reagent (Invitrogen) according to manufacturers' instructions. DF-1 and PGC fibroblast cells were transfected as 293 T cells but with twice the concentration of DNA. Cells were incubated at 37°C. 20–24 hr after transfection, cells were lysed with 50 µl of passive lysis buffer (Promega), and firefly and Renilla luciferase bioluminescence was measured using a Dual-luciferase system (Promega) with a FLUOstar Omega plate reader (BMG Labtech).

## Split luciferase assay

293 T cells were transfected with PB1$^{luc1}$ (25 ng), either PB2 627E or PB2 627K (25 ng), PA (12.5 ng) and chANP32A$^{luc2}$, chANP32AN129I$^{luc2}$, chANP32B$^{luc2}$ or chANP32B$_{33}$$^{luc2}$ (12.5 ng). For split luciferase assays measuring histone interaction, 50 ng of either chANP32A$^{luc2}$, chANP32AN129I$^{luc2}$, H4$^{luc1}$ or H3$^{luc2}$ were transfected into 293 T cells. Control samples assessed the interaction between H4 or PB1$^{luc1}$ and an untagged luc2 construct or the appropriate ANP32A$^{luc2}$ construct and an untagged luc1 construct. All other components transfected into control samples remained consistent with those transfected in with the interacting proteins of interest. 24 hr after transfection, cells were lysed in 50 ul Renilla lysis buffer (Promega) for one hour at room temperature. *Gaussia* luciferase activity was then measured from 10 ul of lysate using the Renilla luciferase kit (Promega) with a FLUOstar Omega plate reader (BMG Labtech). Normalised luminescence ratios were calculated by dividing the luminescence measured from the interacting partners by the sum of the interaction measured from the two controls for each sample (*Figure 6—figure supplement 1*) as previously described (*Cassonnet et al., 2011*).

## Immunoblot analysis

For analysis of PGC derived fibroblasts (*Figure 3b*), at least 300,000 cells were lysed in 60 µl of 1X RIPA lysis buffer (sc-24948, Santa Cruz Biotechnology) according to the manufacturer's instruction. Protein concentration was determined using the Bradford method with the Quick Start Bradford Protein Assay Kit (#5000202, BIORAD) according to the manufacturer's instruction (*Bradford, 1976*) Denaturing electrophoresis and western blotting were performed using the NuPAGE electrophoresis system (Invitrogen) following the manufacturer's protocol. For all other Western blots, cells were lysed in lysis buffer (50 mM Tris-HCl pH 7.8 (Sigma Aldrich), 100 mM NaCl, 50 mM KCl and 0.5% Triton X-100 (Sigma Aldrich), supplemented with cOmplete EDTA free Protease inhibitor cocktail tablet (Roche)) and prepared in Laemmli 2 × buffer (Sigma-Aldrich). Cell proteins were resolved by SDS–PAGE using Mini-PROTEAN TGX Precast Gels (Bio-Rad). Immunoblotting was carried out using the following primary antibodies: rabbit α-ANP32A (Sigma-Aldrich AV40203), mouse α-β-actin (Sigma-Aldrich A2228), mouse α-FLAG (F1804, Sigma-Aldrich), mouse α-Lamin B1 (MAB5492, Merck), mouse α-PCNA (sc-25280, Santa Cruz), rabbit α-Histone 3 (AB1791, Abcam), rabbit α-vinculin (AB129002, Abcam), rabbit α-*Gaussia* Luc (E80235, NEB), rabbit α-PB1 (PA5-34914, Invitrogen) and rabbit α-PB2 (GTX125926, GeneTex). The following secondary antibodies were used: goat anti-rabbit HRP (CST #7074), anti-mouse HRP (CST #7076), goat α-mouse AlexaFluor-568 (A11031, Invitrogen), sheep α-rabbit HRP (AP510P, Merck) and goat α-mouse HRP (STAR117P, AbD Serotec). Protein bands were visualised by chemiluminescence (ECL +western blotting substrate, Pierce) using a FUSION-FX imaging system (Vilber Lourmat).

## Quantification of chANP32A, B and E mRNA levels

Total RNA from PGC fibroblast and DF-1 cells were extracted using an RNeasy mini kit (Qiagen), following manufacturer's instructions. During extraction of RNA, RNeasy columns were treated with RNase-Free DNase (Qiagen). RNA samples were quantified using a Nanodrop Spectrophotometer (Thermo Scientific). Equal concentrations of RNA were subject to first strand synthesis using Revert-tAid (Thermo Scientific) with Oligo(dT) (Thermo Scientific). This product was then quantified with Fast SYBR Green Master Mix (Thermo Scientific) using the following sequence-specific primer pairs: RS17, (5'-ACACCCGTCTGGGCAACGACT-3' and 5'-CCCGCTGGATGCGCTTCATCA-3'), RPL30 (5'-CCAACAACTGTCCTGCTTT-3' and 5'-GAGTCACCTGGGTCAATAA-3'), chANP32A (5'-GTTTGCAACTGAGGCTAAGC-3' and 5'-CAACTGTAGGTCATACGAAGGC-3'), chANP32B (5'-GGTGGCCTTGAAGTTCTAGC-3', and 5'-ATGAGCATCGTCACCTCGC-3'), chANP32E (5'-GAACTAGAGTTTCTTAGCATGG-3' and 5'-TCTCTCTGCAAGGACCTCCAG-3'). Real-time quantitative PCR analysis was performed (Applied Biosystems ViiA 7 Real-Time PCR System).

## Safety/biosecurity

All work with infectious agents was conducted in biosafety level two facilities, approved by the Health and Safety Executive of the UK and in accordance with local rules, at Imperial College London, UK.

## Bioinformatics

ANP32 sequences were downloaded from Ensembl (Gene Trees ENSGT00940000153254 and ENSGT00940000154305.) Amino acid sequences were aligned using MUSCLE (*Edgar, 2004*) and the maximum likelihood tree was constructed using RAxML-HPC2 v.8.2.10 (*Stamatakis, 2014*) (GTRGAMMA model, 100 bootstraps) on XSEDE run on CIPRES (*Miller et al., 2010*). Mapmodulin from *Drosophila melanogaster* was used as an outgroup.

Statistical analysis was performed using GraphPad Prism v.7. Sequencing data was analysed using Geneious Sorfware R6. Image analysis was done using Image J and Microsoft Office 2016.

## Acknowledgements

JSL, CR, SG, MAS, HMS and WSB were supported by Biotechnology and Biological Sciences Research Council (BBSRC) via Strategic LoLa grant BB/K002465/1 'Developing Rapid Responses to Emerging Virus Infections of Poultry (DRREVIP)". AI-A was funded by a Principal's Career Development PhD Scholarship from the University of Edinburgh and PhD funding from Cobb-Vantress, Inc.

BM was supported by the Wellcome Trust. DHG, JS and WSB were supported by grant 205100 from the Wellcome Trust. ES was supported by an Imperial College President's Scholarship. HS was funded from BBSRC grants BB/R007292/1 and BBS/E/I/00007034. HMS and MJM were supported by Institute Strategic Grant Funding from the BBSRC (BB/P013732/1 and BB/P013759/1).

## Additional information

### Competing interests

Jason S Long, Alewo Idoko-Akoh, Michael A Skinner, Helen Sang, Michael J McGrew, Wendy Barclay: A patent application was filed covering the results presented in this article. GB1819200.5. The other authors declare that no competing interests exist.

### Funding

| Funder | Grant reference number | Author |
| --- | --- | --- |
| Biotechnology and Biological Sciences Research Council | BB/K002465/1 | Jason S Long<br>Craig Ross<br>Steve Goodbourn<br>Michael A Skinner<br>Helen Sang<br>Michael J McGrew<br>Wendy Barclay |
| Wellcome | 205100 | Daniel Goldhill<br>Jocelyn Schreyer<br>Wendy Barclay |
| Cobb Vantress | | Alewo Idoko-Akoh |
| Imperial College London | Imperial College President's Scholarship | Ecco Staller |
| University Of Edinburgh | Principal's Career Development | Alewo Idoko-Akoh |
| Wellcome | 105396/Z/14/Z | Bhakti Mistry |
| Biotechnology and Biological Sciences Research Council | BB/R007292/1 | Helen Sang |
| Biotechnology and Biological Sciences Research Council | BBS/E/I/00007034 | Helen Sang |
| Biotechnology and Biological Sciences Research Council | BB/P013732/1 | Helen Sang<br>Michael J McGrew |
| Biotechnology and Biological Sciences Research Council | BB/P013759/1 | Helen Sang |

The funders had no role in study design, data collection and interpretation, or the decision to submit the work for publication.

### Author contributions

Jason S Long, Conceptualization, Data curation, Formal analysis, Supervision, Validation, Investigation, Visualization, Methodology, Writing—original draft, Project administration, Writing—review and editing; Alewo Idoko-Akoh, Conceptualization, Data curation, Formal analysis, Validation, Investigation, Visualization, Methodology, Writing—review and editing; Bhakti Mistry, Conceptualization, Data curation, Formal analysis, Supervision, Validation, Investigation, Visualization, Methodology, Writing—review and editing; Daniel Goldhill, Conceptualization, Data curation, Formal analysis, Investigation, Methodology, Writing—review and editing; Ecco Staller, Data curation, Investigation, Methodology; Jocelyn Schreyer, Data curation, Formal analysis, Validation, Investigation, Methodology; Craig Ross, Data curation, Investigation, Methodology, Writing—review and editing; Steve Goodbourn, Data curation, Formal analysis, Supervision, Funding acquisition, Investigation, Methodology, Project administration, Writing—review and editing; Holly Shelton, Resources, Writing—review and editing; Michael A Skinner, Conceptualization, Funding acquisition, Investigation,

Visualization, Project administration, Writing—review and editing; Helen Sang, Michael J McGrew, Conceptualization, Formal analysis, Supervision, Funding acquisition, Investigation, Visualization, Project administration, Writing—review and editing; Wendy Barclay, Conceptualization, Supervision, Funding acquisition, Visualization, Project administration, Writing—review and editing

### Author ORCIDs
Jason S Long https://orcid.org/0000-0002-0251-6487
Daniel Goldhill http://orcid.org/0000-0003-4597-5963
Ecco Staller http://orcid.org/0000-0002-8443-5559
Michael A Skinner https://orcid.org/0000-0002-0050-4167
Michael J McGrew http://orcid.org/0000-0001-8213-4632
Wendy Barclay https://orcid.org/0000-0002-3948-0895

### Decision letter and Author response
Decision letter https://doi.org/10.7554/eLife.45066.022
Author response https://doi.org/10.7554/eLife.45066.023

## Additional files

### Supplementary files
• Transparent reporting form
DOI: https://doi.org/10.7554/eLife.45066.018

### Data availability
The data used to generate Figure 1 and Figure 1- figure supplement 1 was downloaded from Ensembl (http://www.ensembl.org/Multi/GeneTree/Image?gt=ENSGT00950000182907). Source data for Figures 2, 3, 4, 5, 6 & 7 are available on Dyrad https://dx.doi.org/10.5061/dryad.338t920.

The following dataset was generated:

| Author(s) | Year | Dataset title | Dataset URL | Database and Identifier |
|---|---|---|---|---|
| Long JS, Idoko-Alewo A, Mistry B, Goldhill DH, Staller E, Schreyer J, Ross C, Goodbourn S, Shelton H, Skinner MA, Sang HM | 2019 | Data from: Species specific differences in use of ANP32 proteins by influenza A virus | https://dx.doi.org/10.5061/dryad.338t920 | Dryad Digital Repository, 10.5061/dryad.j1fd7 |

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
