## [Decision Letter]

Thank you for submitting your article "Species specific differences in use of ANP32 proteins by influenza A virus" for consideration by *eLife*. Your article has been reviewed by three peer reviewers, and the evaluation has been overseen by a guest Reviewing Editor and Patricia Wittkopp as the Senior Editor. The following individual involved in review of your submission has agreed to reveal his identity: Benjamin G Hale (Reviewer #1).

The reviewers have discussed the reviews with one another and the Reviewing Editor has drafted this decision to help you prepare a revised submission.

Summary:

Human ANP32A and ANP32B are important polymerase cofactors of influenza viruses. Avian influenza virus relies exclusively on avian ANP32A, thereby defining the host range of influenza virus and is dependent upon a duplication in ANP32A that is unique to birds. In this work the authors could show that human and avian ANP32B are paralogues with a different evolutionary history. Using a series of genomic deletions or mutations in human and avian cells, they show that influenza polymerase is effectively non-functional in human cells lacking both ANP32A and B, whereas deletion of only ANP32A in chicken cells yields the same phenotype. In complementation experiments, the authors could further show that ANP32B cannot stimulate polymerase activity and that only two amino acid variants common to avian ANP32B render it non-functional for the viral polymerase. Intriguingly, viral replication is severely impaired in chicken cells where ANP32A is knocked out, leading the authors to propose that genetic modification of chickens could serve as a bio-control for influenza virus.

Essential revisions:

1) The authors propose that "avian ANP32B and mammalian ANP32B are paralogues: birds have lost the protein orthologous to human ANP32B and eutherian mammals have lost the protein orthologous to avian ANP32B". This is based on the phylogenetic relationships, that suggest that avian ANP32B and mammalian ANP32B shared a common ancestor gene that derived these proteins by duplication, and then one was lost in avian and the other in mammals. If this is the case, it is likely that avian have the remains of a non functional mammalian ANP32B, and mammals the remains of a non functional avian ANP32B (or ANP32C, its orthologue). Are these pseudogenes found in the genome of mammals and birds? The authors indicate in the discussion that this is the case for humans, but no mention in birds is given.

2) The way the manuscript is written and the data presented are a bit misleading. At the end, the story is that ANP32A, ANP32B and ANP32C are paralogues. Only ANP32A and ANP32B support influenza polymerase replication, with some species-specificity depending on polymerase polymorphisms. ANP32B is not found in avian, and the so called avian ANP32B is in reality ANP32C, and therefore, avian species only have ANP32A to support influenza polymerase activity, as they lack ANP32B. The authors should re-write the Abstract and some of the portions of the manuscript to make this point clear.

3) Although is clear the deletion or alteration of chicken ANP32A make chicken cells resistant to optimal replication with influenza virus, the authors cannot exclude that if mutations are introduced in chicken that abrogate the ability of ANP32A to promote virus replication, this result in a selective pressure that select for mutant influenza viruses able to use avian ANP32B or mutated avian ANP32A. This should be discussed.

4) The authors conclude that PGC aKO are "effectively resistant" to influenza virus replication. Yet, Figure 7 shows clear low-level titres that persist throughout the infection and these are noted in the Discussion (paragraph three). A more rigorous assessment is needed to know if these cells support viral replication, both in low MOI multi-cycle and high MOI single-cycle infections. While this could appear to be a semantic distinction, it is critically important for their proposed use of aKO as a bio-control. Any level of viral replication in aKO cells raises the possibility that the virus will adapt, possibly by acquiring the ability to utilise ANP32B. Do viruses recovered at 48hr contain any novel mutations, especially in 7a or 7c-d where titres persist or even increase at later times?

5) The authors propose using aKO in chickens as a means to control influenza virus replication. While a potentially interesting approach, the current data limit confidence in this conclusion. Influenza virus rapidly adapts to diverse host environments, with as little as a single amino acid change in the polymerase conveying new functionality in using different ANP32 proteins. It would seem likely that adaptive variants would arise under these conditions. Moreover, knock-out analysis in mice shows that ANP32A plays an important role in embryogenesis, and that this is more pronounced in the absence of ANP32B (Reilly et al., 2011). Will this also occur in chickens, which naturally lack a true ANP32B paralog and where the newly-named ANP32C is nonfunctional in at least its ability to support IAV polymerases? A more tempered and nuanced discussion of the potential of aKOs is warranted.

---

## [Author Response]

Essential revisions:1) The authors propose that "avian ANP32B and mammalian ANP32B are paralogues: birds have lost the protein orthologous to human ANP32B and eutherian mammals have lost the protein orthologous to avian ANP32B". This is based on the phylogenetic relationships, that suggest that avian ANP32B and mammalian ANP32B shared a common ancestor gene that derived these proteins by duplication, and then one was lost in avian and the other in mammals. If this is the case, it is likely that avian have the remains of a non functional mammalian ANP32B, and mammals the remains of a non functional avian ANP32B (or ANP32C, its orthologue). Are these pseudogenes found in the genome of mammals and birds? The authors indicate in the Discussion that this is the case for humans, but no mention in birds is given.

The presence of both ANP32B and ANP32C in *Xenopus* and orthologous genes in marsupial mammals strongly suggests that both these genes were present in the common ancestor of mammals and birds. Subsequently, ANP32C was lost in the ancestor of placental mammals and ANP32B was lost in the ancestor of birds. The reviewers suggest that the remains of ANP32C should be found in mammals and ANP32B should be found in birds. We have searched more extensively for evidence of these genes. For humans, the synteny surrounding where a gene orthologous to avian ANP32B should be is the same as in marsupials. There is a small amount of evidence that there are remnants of what would have been an ANP32 gene still present in humans: specifically, LRR3 of avian ANP32B maps to the human genome at this locus. However, for mammalian ANP32B in birds, the synteny of the surrounding genes suggests that the gene was lost through a chromosomal rearrangement (e.g. an inversion) as the surrounding genes in birds are very different.

In humans, there is an intronless gene named ANP32C, which is unrelated to *Xenopus* ANP32C and Avian ANP32Bs, but is closely related to ANP32A. To avoid confusion, we no longer refer to a ANP32C group but an Avian ANP32B clade and make clear that ANP32C in humans is unrelated to avian ANP32Bs.

Figure 1—figure supplement 2, the Results (first paragraph) and the Discussion (subsection “Viral replication is abrogated in chicken cells lacking ANP32A”) have been updated.

To clarify for this reviewer, we now included a schematic of ANP32 nomenclature in Author response image 1.

2) The way the manuscript is written and the data presented are a bit misleading. At the end, the story is that ANP32A, ANP32B and ANP32C are paralogues. Only ANP32A and ANP32B support influenza polymerase replication, with some species-specificity depending on polymerase polymorphisms. ANP32B is not found in avian, and the so called avian ANP32B is in reality ANP32C, and therefore, avian species only have ANP32A to support influenza polymerase activity, as they lack ANP32B. The authors should re-write the Abstract and some of the portions of the manuscript to make this point clear.

The orthology of ANP32 proteins is clarified from point 1. To further clarify, the Abstract has been reworded and the focus is on the 129I residue as the key difference that occurs only in birds and not other members of the clade.

3) Although is clear the deletion or alteration of chicken ANP32A make chicken cells resistant to optimal replication with influenza virus, the authors cannot exclude that if mutations are introduced in chicken that abrogate the ability of ANP32A to promote virus replication, this result in a selective pressure that select for mutant influenza viruses able to use avian ANP32B or mutated avian ANP32A. This should be discussed.

We agree with the reviewers deductions. In future we plan to examine the evolution of virus in cells lacking or with gene edited ANP32A.

We have acknowledged this possibility in the Discussion paragraph three.

4) The authors conclude that PGC aKO are "effectively resistant" to influenza virus replication. Yet, Figure 7 shows clear low-level titres that persist throughout the infection and these are noted in the Discussion (paragraph three). A more rigorous assessment is needed to know if these cells support viral replication, both in low MOI multi-cycle and high MOI single-cycle infections. While this could appear to be a semantic distinction, it is critically important for their proposed use of aKO as a bio-control. Any level of viral replication in aKO cells raises the possibility that the virus will adapt, possibly by acquiring the ability to utilise ANP32B. Do viruses recovered at 48hr contain any novel mutations, especially in 7a or 7c-d where titres persist or even increase at later times?

We agree with the authors observations and acknowledge that to describe the aKO PGCs as ‘resistant’ is misleading. We have altered this to ‘resilient’ in the manuscript.

We have repeated the virology in the WT and aKO PGC fibroblast cells in order to robustly test their resilience to infection. Previously we did not pH wash cells to remove any virus present in the inoculum for fear of damaging the PGC fibroblasts. Since then we have found they tolerate it well and have disregarded previous virus growth curves as we could not be confident if we were measuring residual virus from the inoculum or true virus amplification. This has been updated in the Materials and methods.

We infected WT and aKO PGC fibroblasts with either PR8 reassortant H5N1 or H7N9 virus at an MOI of 0.0001 or 1.0. We observed no or very limited replication in the aKO cells at the lower MOI and were unable to sequence from this material at the 48 hour time point. However, we observed attenuated replication in aKO cells at the higher MOI.

We sequenced the PB2 gene from the H5N1 and Anhui virus 24hr time point in the high MOI conditions and found the PB2 sequence of viruses from aKO cells to be identical to those from WT cells and the virus input.

It remains unclear how the virus may replicate in the absence of ANP32A at higher MOIs when polymerase activity in the minigenome assay is abrogated and this raises interesting questions to pursue in future experiments.

Changes to the manuscript include the Results, Materials and methods, Discussion and a new version of Figure 7.

5) The authors propose using aKO in chickens as a means to control influenza virus replication. While a potentially interesting approach, the current data limit confidence in this conclusion. Influenza virus rapidly adapts to diverse host environments, with as little as a single amino acid change in the polymerase conveying new functionality in using different ANP32 proteins. It would seem likely that adaptive variants would arise under these conditions. Moreover, knock-out analysis in mice shows that ANP32A plays an important role in embryogenesis, and that this is more pronounced in the absence of ANP32B (Reilly et al., 2011). Will this also occur in chickens, which naturally lack a true ANP32B paralog and where the newly-named ANP32C is nonfunctional in at least its ability to support IAV polymerases? A more tempered and nuanced discussion of the potential of aKOs is warranted.

We understand the reviewer’s concern that we are over-stating the impact of our research. We only state here that our data suggest this as a possible and very novel approach. We believe that the data here warrant the generation of ANP32A gene-edited chickens to test this possibility as a next step and we have recently obtained funding to this end. Part of this project will include the examination of virus evolution in conditions where ANP32A is lacking or altered.

We have better represented the limitations of our findings in the Discussion to acknowledge the possible detrimental effect on host of ANP32A modification and discuss the need to examine virus evolution in the future.